Trapdoor proof of work

http://orcid.org/0000-0002-0907-3252 Capocasale Vittorio vittorio.capocasale@polito.it
Polytechnic Institute of Turin , Turin , Italy
Shah Rahul
Electronic publication date: 2024 Jan 19
Publication date: 2024
Volume: 10
Electronic Location ID: e1815
Received 2023 Jun 26; Accepted 2023 Dec 19
Copyright: © 2024 Capocasale
Copyright year: 2024
Copyright holder: Capocasale
License: This is an open access article distributed under the terms of the Creative Commons Attribution License, which permits unrestricted use, distribution, reproduction and adaptation in any medium and for any purpose provided that it is properly attributed. For attribution, the original author(s), title, publication source (PeerJ Computer Science) and either DOI or URL of the article must be cited.
License URL: https://creativecommons.org/licenses/by/4.0/

Keywords: Blockchain, Nakamoto consensus, Proof of work, Chameleon hash function

Funding: The authors received no funding for this work.

==============================
Consensus algorithms play a crucial role in facilitating decision-making among a group of entities. In certain scenarios, some entities may attempt to hinder the consensus process, necessitating the use of Byzantine fault-tolerant consensus algorithms. Conversely, in scenarios where entities trust each other, more efficient crash fault-tolerant consensus algorithms can be employed. This study proposes an efficient consensus algorithm for an intermediate scenario that is both frequent and underexplored, involving a combination of non-trusting entities and a trusted entity. In particular, this study introduces a novel mining algorithm, based on chameleon hash functions, for the Nakamoto consensus. The resulting algorithm enables the trusted entity to generate tens of thousands blocks per second even on devices with low energy consumption, like personal laptops. This algorithm holds promise for use in centralized systems that require temporary decentralization, such as the creation of central bank digital currencies where service availability is of utmost importance.

Introduction

Peer-to-peer protocols, such as the blockchain (Nakamoto, 2008) and the interplanetary file system (Capocasale, Musso & Perboli, 2022), constitute the core technologies of the Web 3.0 revolution (Alabdulwahhab, 2018). By leveraging blockchain technology, a group of peers can manage a tamper-resistant and decentralized ledger, ensuring transparent and secure data sharing (Hellani et al., 2021). These properties are particularly valuable for decision-makers, as fairness and optimality are common requirements in decision processes (Aringhieri et al., 2022; Diglio et al., 2021; Sale et al., 2018), making blockchain applicable in various sectors (Elia et al., 2022; Khan & Anjum, 2022; Hasan et al., 2022). However, according to the scalability trilemma conjecture, a blockchain cannot simultaneously achieve security, decentralization, and scalability (Perboli, Musso & Rosano, 2018). The intuitive reason for such a constraint lies in the fact that each peer in a blockchain network does not trust the others, keeps a full copy of the ledger, and autonomously verifies each transaction. Thus, instead of distributing workloads, blockchain systems replicate them on each peer.

Consensus algorithms play a fundamental role in blockchain systems by enabling the replication of the ledger’s state across multiple peers (Antoniadis et al., 2018). Consensus algorithms are one of the main causes of the scalability trilemma (Altarawneh et al., 2020), as the replication process involves dealing with various variables, such as network delays and the presence of selfish or malicious peers.

In general, consensus algorithms can efficiently guarantee state synchronization in the presence of crashing peers (Crash Fault-Tolerant (CFT) consensus algorithms). However, when malicious peers need to be taken into account more resilient algorithms must be used (Byzantine Fault-Tolerant (BFT) consensus algorithms). However, such algorithms sacrifice efficiency for robustness (Du et al., 2017; Capocasale, Gotta & Perboli, 2023). Consequently, in environments with high-efficiency requirements, only CFT algorithms are viable (Gatteschi et al., 2018).

In a decentralized scenario, it is not possible to guarantee the absence of malicious peers. As a result, numerous studies in the literature have attempted to improve the efficiency of existing BFT consensus algorithms (Pass & Shi, 2017; Wu, Song & Wang, 2020; Xu et al., 2019; Zhou, Hua & Jin, 2020; Abuidris et al., 2021; Liu, Tan & Zhuo, 2022).

However, in certain scenarios, systems consist of both honest parties and potentially malicious peers. In such situations, CFT algorithms are unsuitable as they cannot tolerate malicious peers. Conversely, BFT algorithms sacrifice efficiency unnecessarily by protecting the system from honest parties. Therefore, even efficient BFT algorithms are sub-optimal. We propose a CFT consensus algorithm that allows peers to delegate decisions to honest parties and switches to BFT when honest parties crash. This dynamic approach ensures efficiency is only sacrificed when necessary without compromising the security of the system.

Given that the existing literature does not consider the possibility of an algorithm that dynamically and automatically switches between CFT and BFT, and that hybrid approaches between CFT and BFT algorithms have not been explored, we offer the following contributions: We introduce Trapdoor Proof of Work (TPoW), which combines Proof of Work (PoW) (Nakamoto, 2008) and chameleon hash functions (Ateniese et al., 2017). TPoW is a mining algorithm that can replace PoW in Nakamoto consensus protocols, allowing a group of entities to make decisions efficiently by relying on a trusted party: the trusted party can quickly solve the TPoW challenge and exploit a trapdoor to minimize energy waste. Moreover, when the trusted party cannot participate in the consensus, the remaining entities can still make decisions by relying on a fully decentralized but less efficient protocol.

We conduct the first experimental validation of TPoW-based Nakamoto consensus.

We describe potential applications of TPoW-based Nakamoto consensus.

The remaining part of this article is structured as follows: “Background” introduces the main concepts related to blockchain, consensus algorithms, and chameleon hash functions; “Trapdoor Proof of Work” describes TPoW; “Protocol Analysis” analyzes TPoW; “Experimental Validation and Discussion” presents the experimental validation of TPoW and the related discussion; “Conclusion” concludes the article.

Background

This section provides a brief overview of the main concepts relevant to this study. Additionally, it introduces the problem under investigation and summarizes the key solutions proposed in the literature.

Chameleon hash functions

A chameleon hash function is a cryptographic hash function that incorporates a trapdoor, allowing for the discovery of arbitrary collisions (Krawczyk & Rabin, 1998). Without knowledge of the trapdoor, chameleon hash functions are equivalent to regular cryptographic hash functions (Khalili, Dakhilalian & Susilo, 2020). This study utilizes private-coin chameleon hash functions (Ateniese et al., 2017). A private-coin chameleon hash function consists of the following algorithms (Ateniese et al., 2017): (hk, tk) ← CHGen(1κ) is a probabilistic algorithm that generates the hash key ( hk) and the trapdoor key ( tk).

(h, ξ) ← CHash(hk, m;r) is a probabilistic algorithm that hashes the message m using private-coin randomness r∈Rhash. CHash produces the chameleon hash ( h) and the check value ( ξ).

d ← CHVer(hk, h, m, ξ) is a deterministic algorithm that outputs true if the hash h with the check value ξ is a valid chameleon hash for the message m. CHVer returns false otherwise.

ξ′ ← CHCol(tk, (h, m, ξ), m′) is a probabilistic algorithm that enables the discovery of hash collisions. Specifically, given a message m′ and a valid tuple (h,m,ξ), CHCol outputs ξ′ such that (h,m′,ξ′) remains a valid tuple. In other words, CHVer(hk, h, m′, ξ′) returns true.

Blockchain

A blockchain is a distributed ledger managed by a group of peers, where data is organized into blocks. Each block consists of a body that contains ledger entries (transactions) and a header that contains metadata (Nakamoto, 2008). Notably, a block’s header includes the hash of the previous block’s header.

In this study, the notation for blocks is as follows:

B:=(sB,xB,hB,ξB),

where sB∈0,1κ represents the cryptographic hash of the previous block, xB∈0,1∗ denotes the list of transactions included in the current block (i.e., the body), and hB and ξB denote the chameleon hash of the block and its check value. Figure 1 visually illustrates the blockchain structure employed in this study.

Figure 1 Structure of a TPoW blockchain.

Each block stores the cryptographic hash of its predecessor, preserving immutability. However, the nodes utilize the chameleon hash of the block for mining.

It is worth noting that, within the context of this study, the term centralized is the opposite of decentralized, not distributed. A system is considered distributed if it consists of physically separated modules, while it is deemed decentralized if it is managed by multiple entities (Capocasale, Gotta & Perboli, 2023).

Consensus algorithms

A consensus algorithm is a protocol that enables a set of nodes to reach a mutual agreement on a decision, where all possible outcomes of the decision are equally beneficial. In a typical scenario, the nodes are geographically dispersed and can only communicate through message exchange. However, messages may experience delays, loss, or arrive out of order. In extreme cases, certain nodes may even send misleading messages to hinder the establishment of a common agreement. The objective of a consensus algorithm is to ensure that the non-faulty nodes reach the same decision. Therefore, consensus algorithms must consider a fault-tolerance model and a network model.

In terms of the fault-tolerance model, consensus algorithms that tolerate the failure of some nodes are referred to as Crash Fault-Tolerant (CFT) algorithms. Those that can also handle the presence of malicious nodes are known as Byzantine Fault-Tolerant (BFT) algorithms (Baliga, 2017).

Regarding the network model, consensus algorithms assume the existence of an invisible adversary who controls network delays, including message ordering. Synchronous model: Messages experience delays of at most a finite and known Δ.

Asynchronous model: Messages experience delays of at most a finite but unknown Δ.

Partially-synchronous model: The invisible adversary triggers a Global Stabilization Time (GST) event at an unknown moment. The network operates asynchronously before GST (during a transitory phase) and synchronously after GST (during the normal phase).

Consensus algorithms possess two fundamental properties (Baliga, 2017). Safety: All (honest) participants reach the same decision.

Liveness: A decision is reached within a finite amount of time.

Proof of Work (PoW) (Back, 2002) is a protocol designed to prevent denial-of-service attacks. It was later adapted as a Sybil protection technique for Nakamoto consensus (Nakamoto, 2008). PoW involves a computationally intensive challenge that can only be solved through brute-force computation. In the context of blockchain, a block can be added to the chain if the hash of its header falls within a predefined threshold. Specifically, a valid block can be generated by iteratively modifying a portion of the header known as the nonce.

When multiple peers solve the challenge almost simultaneously, they may add different blocks to their respective chains. This situation leads to a fork, where multiple branches coexist. As one branch grows longer than the others, peers switch to it. A branch can outgrow others in the same period only if more peers support it. Thus, this protocol incorporates a majority-based voting scheme, enabling peers to reach a consensus. This consensus algorithm is known as Nakamoto consensus and combines PoW with the longest chain rule. In the context of Nakamoto consensus, safety and liveness are defined by the following properties (Pass, Seeman & Shelat, 2017). Consistency: The chains of two honest peers can differ only in the last T blocks with overwhelming probability in T.

Future self-consistence: At any two points r and s, the chains of any honest player at r and s differ only within the last T blocks with overwhelming probability in T.

g-chain growth: At any point in the execution, the chain of honest players grows by at least T transactions in the last Tg rounds with overwhelming probability in T.

μ-chain quality: at least μ of any T consecutive transactions in any chain held by some honest player were submitted by honest players with overwhelming probability in T.

Erdős–rényi random graphs

In the field of graph theory, a graph is considered connected if there exists a path between any two nodes. An Erdős–Rényi random graph, denoted as G(n,p), is an undirected graph with n nodes, where each edge has a probability p of being present. An Erdős–Rényi random graph is said to be almost surely connected for some ϵ>0 if the following condition holds (Erdős & Rényi, 1960):

(1) p>(1+ϵ)ln⁡nn.

Problem statement

Currently, consensus algorithms can be used in the following scenarios: if there are no malicious peers, both Crash Fault-Tolerant (CFT) and Byzantine Fault-Tolerant (BFT) algorithms are applicable. However, CFT algorithms tend to be more efficient. If peers are potentially malicious, only BFT algorithms can be used, but they sacrifice efficiency for robustness.

However, real-world applications may present scenarios that fall in between. These scenarios involve a mix of trustworthy parties and potentially malicious peers within the system. Therefore, the problem under consideration is finding consensus within a group of nodes that includes both peers, which are nodes that may try to disrupt the consensus process, and trusted parties, which we define as nodes that are guaranteed to follow the consensus protocol’s rules and do not indulge in selfish, colluding, or disruptive behaviors. For instance, parties with economic, regulatory, or reputation-driven interests might be considered trustworthy. CFT algorithms are not suitable for solving this problem since they cannot tolerate malicious peers. On the other hand, BFT algorithms sacrifice efficiency unnecessarily because they protect the system from trusted parties that would never behave maliciously. Therefore, even efficient BFT algorithms are suboptimal. To address this, we have designed a CFT consensus algorithm that allows peers to delegate decisions to trusted parties and automatically switches to a BFT algorithm when those trusted parties are offline. This dynamic approach ensures efficiency is sacrificed only when necessary, without compromising the security of the system. We believe such an approach could find adoption in centralized systems that need to transition to a decentralized state for short periods.

Related work

The main ideas behind this study draw from Proof of Work (PoW) (Back, 2002) and private-coin chameleon hash functions (Ateniese et al., 2017). The combination of blockchain and chameleon hash functions was proposed in Ref. (Ateniese et al., 2017) and further explored in various studies (Ashritha, Sindhu & Lakshmy, 2019; Huang et al., 2020; Precht & Marx Gómez, 2020; Wu, Ke & Du, 2021; Jia et al., 2022). However, the proposed solution aimed to create a redactable blockchain rather than a consensus algorithm.

Nakamoto consensus has high energy consumption (Mardiansyah & Sari, 2021), which pushed several authors to propose modifications to reduce it. Proof of Elapsed Time (PoET) (Chen et al., 2017) utilizes hardware components to guarantee Byzantine fault tolerance. However, the security of such components has been questioned (Schwarz, Weiser & Gruss, 2019). Proof of Stake (PoS) algorithms (e.g., Ouroboros (Kiayias et al., 2017)) assume that nodes are rational agents acting to maximize their economic return. Therefore, PoS algorithms are not usable without a cryptocurrency with real economic value. Some authors have proposed combining multiple mining algorithms. Proof of Contribution (Xue et al., 2018) is an example of such an approach, which combines PoW and PoS.

Modifications to the longest chain rule have also been explored. IOTA (Popov, 2018) utilizes a directed acyclic graph (DAG) instead of a linear sequence of blocks to represent the ledger. The Tangle 2.0 Leaderless Nakamoto Consensus (Müller et al., 2022) employs the heaviest DAG rule and a stake- or reputation-based weight function to reach an agreement.

Liu et al. (2016) introduces XPaxos, the first Cross Fault Tolerance (XFT) consensus algorithm. XFT algorithms offer stronger consistency and availability guarantees than CFT algorithms. Moreover, XFT algorithms provide stronger availability guarantees than BFT algorithms. However, XFT algorithms are applicable only when Byzantine nodes are not coordinated.

Hybrid consensus algorithms, where existing BFT protocols are combined to offer different compromises between efficiency and decentralization, have been explored in many studies (Pass & Shi, 2017; Wu, Song & Wang, 2020; Xu et al., 2019; Zhou, Hua & Jin, 2020; Abuidris et al., 2021; Liu, Tan & Zhuo, 2022). However, none of these proposals specifically focuses on a consensus algorithm that dynamically and automatically switches between CFT and BFT protocols. This study addresses that gap by describing the general idea and providing possible applications. Table 1 provides a summary of the literature.

Table 1 Contextualization of this study within the existing literature.

Algorithm	Key ideas	Advantages	Disadvantages	
Proof of work (Nakamoto)	Permissionless (no identities), eventual safety	Secure, scalable	Not efficient, energy-hungry	
Proof of elapsed time (Nakamoto)	Usage of trusted computing devices	Low energy consumption	Vulnerabilities in trusted computing components	
Proof of stake (Nakamoto)	Game theory, rational agents	Energy savings, improved efficiency	Requires currency	
Proof of contribution (Nakamoto)	PoW + PoS	Inherited from PoW and PoS	Inherited from PoW and PoS	
Leaderless Nakamoto consensus	No leader election, optimisitc and concurrent block creation	Extension to DAG	Requires currency or reputation	
xPaxos	All-mighty adversaries are unlikely	Consistency and availability guarantees	Cannot tolerate all-mighty adversaries	
Hybrid algorithms	Two BFT algorithms combined	Compromises between the original algorithms	Compromises between the original algorithms	
This study (Nakamoto)	Trusted parties are common	Enhanced efficiency, potential energy savings	Trusted party needed to have advantages	

Trapdoor proof of work

This section provides an overview of Trapdoor Proof of Work (TPoW).

Protocol overview

TPoW is based on the original Proof of Work (PoW) implementation introduced by Bitcoin (Lánskỳ, 2017). However, in TPoW, the pre-image challenge is built upon chameleon hash functions instead of cryptographic hash functions.

Our protocol uses a cryptographic hash function to establish the linkage between each block and its predecessor, while a chameleon hash function is employed for the PoW computation (Fig. 1). The security and resilience of TPoW heavily rely on the strength of the underlying chameleon hash functions. It’s important to note that certain implementations of chameleon hash functions have encountered key-exposure issues (Ateniese & Medeiros, 2004). However, for the purpose of this study, theoretical chameleon hash functions are assumed to exist.

In TPoW, a block is considered valid if its chameleon hash is sufficiently close to a predetermined value, which corresponds to the chameleon hash of the genesis block. The nodes within the network can approach the pre-image challenge through two methods (Fig. 2): brute-force computation or by utilizing the trapdoor knowledge. The brute-force method involves an inefficient trial-and-error approach, whereas the trapdoor method enables an efficient and direct computation. Consequently, any node possessing the trapdoor can effectively solve the TPoW challenge almost instantaneously. However, for nodes without the trapdoor knowledge, the challenge remains as difficult as the original PoW.

Figure 2 A flowchart describing the generation of a block through the TPoW protocol.

Algorithms 1–4 summarize the primary steps involved in the TPoW protocol.

Algorithm 1 Genesis generation algorithm.

1: function Genesis (κ,D)▹ Genesis receives the security parameter (κ) and the difficulty paremeter (D)	
2:    (hk,tk)← CHGen(1κ)                        ▹Generation of the chameleon hash keys	
3:    sT←0                          ▹ The hash of the previous block is zero by convention	
4:    xT←(hk,D,κ)▹ For simplicity, the TPoW difficulty (D), hk, and κ are published in the body of the genesis	
5:    mT←Hash (sT, xT)                         ▹ Hash is a cryptographic hash function	
6:    rT←RndExtract()      ▹ Private randomness rT is randomly extracted from its domain	
7:   (hT, ξT)←CHash(hk, mT, rT)  ▹ Generation of the chameleon hash and the check value of the genesis	
8:    T←(sT,xT,hT,ξT)                                    ▹ Generation of the genesis block	
9:   return T	
10: end function	

Algorithm 2 Block validity verification algorithm.

1: function Verify(A, B, T)▹ Verify receives the current chain head (A), the block to append to it (B), and the genesis (T)	
2:   mA←Hash(A.s, A.x)               ▹ The cryptographic hash of the current chain head	
3:   mB←Hash(B.s, B.x)                     ▹ The cryptographic hash of the new block	
4:   p1←Equal(B.s, mA)     ▹ A valid block contains the cryptographic hash of its predecessor	
5:   p2 ← CHVer(hk, B.h, mB, B.ξ)          ▹A valid block must have a valid chameleon hash	
6:   p3 ← Diff(T.h, B.h) 2κ<2κD▹ The chameleon hash of a valid block must be close enough to the chameleon hash of the genesis. Diff performs the binary subtraction between T.h and B.h and converts the result into an integer. D and κ can be retrieved from the genesis block	
7:   return p1 AND p2 AND p3                     ▹ A valid block satisfies all the previous conditions	
8: end function	

Algorithm 3 Slow block mining algorithm (it does not require knowledge of tk).

1: function Mine _Slow(A, T)▹ Mine _Slow receives the current chain head (A) and the genesis (T)	
2:    xB← Fill_Transactions() ▹ Fill_Transaction() includes some transactions in the block	
3:   mA ← Hash(A.s, A.x)                                 ▹ The cryptographic hash of A	
4:   mB ← Hash(mA, xB)                                   ▹ The cryptographic hash of B	
5:   repeat	
6:     rB ← RndExtract() ▹ The private randomness rB is randomly extracted from its domain	
7:      (hB, ξB) ← CHash(hk, mB, rB)          ▹ The chameleon hash is computed with the current randomness	
8:      B←(mA,xB,hB,ξB)                                 ▹ The candidate block B is generated}	
9:   until Verify(A, B, T)                               ▹The validity of B must be checked	
10:  return B	
11: end Function	

Algorithm 4 Fast block mining algorithm (it requires knowledge of tk)

1:  function Mine _Fast(A, T, tk)  ▹ Mine _Fast receives the current chain head (A), the genesis (T), and tk	
2:     xB←Fill_Transactions () ▹ Fill_Transaction includes some transactions in the block	
3:     mA ← Hash(A.s, A.x)                                 ▹ The cryptographic hash of A	
4:    mB ← Hash(mA, xB)                                  ▹ The cryptographic hash of B	
5:     mT ← Hash(T.s, T.x)                                 ▹ The cryptographic hash of T	
6:     ξB ← CHCol(tk, (T.h, mT, T.ξ), mB)▹ Hash collision: (T.h,mT,T.ξ) and (T.h,mB,ξB) are both valid tuples	
7:     B←(mA,xB,T.h,ξB)                                         ▹ A valid block is generated	
8:    return B	
9: end function	

Safety and liveness

We provide simple proof of the safety and liveness of TPoW by leveraging well-known PoW-related results. The following theorem is proven in a previous work (Pass, Seeman & Shelat, 2017).

Theorem 1. “Assume ρ<12. Then for every n, Δ, there exists some sufficiently small p0=Θ(1Δn) such that Nakamoto’s protocol with mining parameter p≤p0 satisfies consistency, future self consistency, 1−ρ1−ρ‒chain quality, and pn2‒growth”.

In particular, let ρ represent the fraction of computational power controlled by malicious nodes, n denote the total number of nodes, Δ be the time delay parameter defining the network model, and p0 indicate the mining hardness parameter, which represents the probability of finding a valid nonce in a single attempt.

From a practical perspective, the theorem establishes the consistency and liveness properties of the Nakamoto consensus if the computational power controlled by malicious nodes is less than half of the total computational power of the network.

We can prove the following corollary:

Corollary 1. Assuming ρ<12 and the existence of a Nakamoto protocol with a mining parameter p satisfying Theorem 1, there exists a TPoW-based Nakamoto protocol with a mining parameter p satisfying Theorem 1.

Proof. We begin with a Nakamoto protocol that satisfies Theorem 1. We replace the cryptographic hash function used in PoW with a chameleon hash function that has the same mining parameter. The trapdoor knowledge is not provided to any of the peers. Under these conditions, cryptographic hash functions and chameleon hash functions are equivalent by definition (Khalili, Dakhilalian & Susilo, 2020). Therefore, the modified protocol still satisfies Theorem 1. Next, we introduce a trusted party into the system and provide the trapdoor knowledge exclusively to this trusted party. Since the trusted party is honest by definition, we are only increasing the mining power of the honest peers. Thus, the assumption ρ<12 is not violated, and the protocol continues to satisfy Theorem 1. Moreover, the protocol we have constructed is precisely a TPoW-based Nakamoto consensus. Therefore, the thesis is proven by construction.

Protocol analysis

The analysis of TPoW covers various aspects, taking into account a conservative uptime ratio of 90% for the trusted party. It is worth noting that typical production systems achieve even higher uptime ratios of 99% or more (Cérin et al., 2014). However, in markets with high volatility or applications with critical missions, even brief periods of service downtime can have severe consequences, as discussed in “Possible Applications”.

Immutability

As depicted in Fig. 1, TPoW blocks are linked using a cryptographic hash function. Therefore, past blocks cannot be modified without altering all subsequent blocks. However, the chameleon hash and the check value can be changed as long as Algorithm 2 returns true. This flexibility is not problematic because the chameleon hash and the check value do not impact the ledger’s state or the chain of hashes. Nonetheless, storing the chameleon hash and the check value in the following block can prevent their alteration.

Finality

Even though the chain is immutable, forks can still occur. When the trusted party is online, transaction consolidation is guaranteed after a single block if the difficulty parameter is appropriately configured. In fact, the trusted party can publish blocks faster than the combined peers and should have no incentive to fork the system. When the trusted party is offline, the TPoW-based Nakamoto consensus has the same level of finality as the PoW-based consensus. The blocks generated by the trusted party are identifiable since they have the same chameleon hash as the genesis block. Thus, peers are aware of the number of block confirmations necessary to finalize a given transaction.

Message complexity

If the identity of the trusted party is public, each peer can directly query the trusted party to obtain the latest block. In the absence of the trusted party, each peer should query other peers and rely on the longest chain among the responses received. If the network is connected, querying a single honest peer is sufficient to receive the longest chain since such a chain is formed by the honest majority. Hence, we model the network as an Erdős Rényi random graph G(n,p) where each node queries k other peers. Additionally, we assume that q out of the n nodes are faulty. Only edges connecting honest peers need to be considered since faulty nodes may never respond to queries. Thus, G is almost surely connected if:

(2) kn(n−qn)2>(1+ϵ)ln⁡nn.

For instance, if n=106 and q=n3, the honest nodes should randomly query k=32 other peers to obtain the longest chain.

Therefore, the message complexity of the TPoW-based Nakamoto consensus is O(kn). Specifically, k=1 when peers query the trusted party directly, and k=32 when peers query each other. Taking into account the assumed uptime ratio of the trusted party, the average value for k is 4. By comparison, under normal conditions, the message complexity is O(n) for Raft (Ongaro & Ousterhout, 2019), O(n2) for PBFT (Castro & Liskov, 1999), O(kn) with k=32 for PoW-based Nakamoto consensus (as per the previous analysis), and O(kn) for Avalanche (where k=20 is currently used) (Rocket et al., 2019).

Energy efficiency

TPoW could result in energy savings since the trusted party generates most of the blocks most of the time. For example, finding a 256-bit hash collision on a personal laptop takes about 165±5 microseconds1 . In contrast, the thousands of miners in the Bitcoin network require approximately 10 minutes to find a 46-bits hash collision, according to current statistics (Blockchain.Com, 2022). Therefore, attempting to solve the TPoW challenge is not advantageous for peers when the trusted party is online. Instead, the optimal strategy for peers is to wait for some time before addressing the TPoW challenge. This waiting period can be estimated based on the average chain growth rate. As a result, energy is only wasted when the trusted party is offline, potentially leading to energy savings.

Horizontal scaling

Since no single peer is likely to solve the PoW challenge two or more times consecutively, PoW does not allow for parallel block production. In contrast, TPoW enables horizontal scaling as the trusted party can optimistically create blocks in parallel and chain them afterward. The only limitation lies in the possible read/write conflicts among different transactions. Transactions can declare their read/write sets to overcome these limitations, and a parallel scheduler can order them. Hyperledger Sawtooth (Olson et al., 2018) adopts this approach.

Properties

Based on the previous discussion, the TPoW protocol possesses several features: It does not rely on public key infrastructure or assumptions regarding clock synchronization.

It is efficient since the trusted party can leverage the trapdoor to quickly solve the TPoW challenge without wasting energy.

It is resilient as it does not depend on the active participation of the trusted party. The TPoW challenge remains solvable even without knowledge of the trapdoor. In such cases, TPoW behaves like a standard PoW.

It is dynamic and automatic, with each node operating independently of others. The presence of the trusted party does not need to be notified to peers to take advantage of improved efficiency, and no action is required when the trusted party ceases to participate in the protocol. The transition between BFT and CFT is automatic, unnoticeable, and does not introduce additional risks.

The trusted party has complete control over the blockchain. The trusted party can solve the TPoW challenge faster than all other parties combined. Consequently, the trusted party can fork the system at any time by producing a branch longer than the currently active one (similar to a 51% attack). However, the trusted party is honest by definition and should only fork the system in exceptional situations such as stolen assets. To prevent deep forks, finality gadgets could be added to the TPoW protocol (Buterin & Griffith, 2017).

Experimental validation and discussion

This section describes and discusses the experimental tests performed to validate some of the theoretical results obtained in “Protocol Analysis”.

Test description

We created a TPoW simulation based on the implementation of chameleon hash functions provided in Ref. (Willems, 2020). We measured the number of blocks mined by a varying number of processes in two minutes. All processes pushed the blocks to a shared data structure protected by a mutex. A process could be a peer or a trusted party. The difference between the two lies in the fact that peers are not aware of the trapdoor and cannot efficiently find collisions, whereas trusted parties can. We examined the behavior of TPoW both when a trusted party was online and when it was not. We also examined the behavior of TPoW for different values of the difficulty parameter. We conducted the tests on an Acer Nitro AN515-44 laptop and used the Go programming language version 1.16.

Discussion

The results of the tests are summarized in Fig. 3 and confirm the theoretical analysis of TPoW. Figure 3A shows the behavior of TPoW when no trusted party is online. As expected, increasing the number of peers also increases the number of mined blocks if the difficulty is kept constant, in conformity with PoW. Moreover, when the difficulty parameter increases from 28 to 216 and the number of peers is kept constant, the number of mined blocks decreases by approximately a factor of 28. This indicates that when no trusted party is online, TPoW behaves similarly to PoW, and the difficulty parameter can be adjusted to suit the network size, for example, to meet a target block period, as in the case of Bitcoin.

Figure 3 Performance evaluation of the TPoW-based Nakamoto consensus.

Conversely, when the trusted party is online (Fig. 3B), the difficulty parameter does not affect the mining process. The lines representing the number of mined blocks by the trusted party with different difficulties overlap almost perfectly, suggesting that the trusted party can mine blocks at a high rate regardless of the difficulty. This result confirms that TPoW is energy-efficient when the trusted party is online, and the trusted party can generate more blocks than all the combined peers (as depicted in Figs. 3C and 3D).

An unexpected finding during the tests was that the number of mined blocks decreased as the number of peers increased. This behavior was attributed to the current testing strategy, which involves a shared data structure protected by a mutex. The acquisition of the mutex likely becomes more time-consuming as the number of contenders increases, causing a slowdown for the trusted party. Thus, improving the testing strategy could lead to obtaining the expected behavior.

Figure 4 shows a comparison of the blocks mined by a single node, either a TPoW peer, a TPoW trusted party, or a PoW peer. Depending on the difficulty, the TPoW trusted party can outperform a PoW peer by several orders of magnitude. Upon comparing the two types of peers, it is evident that the TPoW peer produces fewer blocks, possibly due to the complexity of computing chameleon hashes. However, in real-world scenarios, this does not translate to a performance loss but rather to the use of simpler challenges. In fact, the difficulty parameter is usually tuned to obtain a target block period, which must be as short as possible but also much longer than the block propagation time to guarantee safety (Pass, Seeman & Shelat, 2017). This restriction does not apply to the trusted party, which is assumed to be honest and should produce blocks at the highest rate possible. The results of this test further confirm that TPoW behaves like PoW when the trusted party is offline.

Figure 4 Comparison between TPoW and PoW.

Possible applications

TPoW could simplify network management in certain use cases. We believe that TPoW could find adoption in systems that need to become decentralized for short periods. We propose two scenarios: enhanced availability and gradual decentralization. However, other scenarios are likely to emerge in the future.

Service availability is a key factor in mission-critical or financial applications. For example, periods of market volatility are characterized by the sudden generation of high volumes of transactions, particularly when investors use automated trading tools (Attanasio et al., 2019). However, centralized exchanges cannot process such a high volume of transactions as they usually handle much lower volumes, leading to their services going offline. Consequently, investors are unable to submit transactions when they need to. One such situation occurred in May 2021, with investors losing millions of U.S. dollars in a cryptocurrency market crash (Kowsmann & Ostroff, 2021).

If a user relies on the services of a centralized exchange, they must trust the company managing it. Thus, TPoW could be exploited by centralized exchanges to decentralize their architecture when their systems are overwhelmed by users’ requests. Such an approach would safeguard the availability of their services. Similarly, TPoW could enable the development of resilient central bank digital currencies (CBDCs) (Benedetti et al., 2022): users would be able to transact even when the trusted central bank is offline.

In recent years, smart contracts (Capocasale & Perboli, 2022) have created the opportunity to transform centralized services into decentralized ones (Hribernik et al., 2020; Serrano, 2022). However, such transformations are often experimental, as decentralized paradigms are not yet well-established. To mitigate decentralization uncertainty, projects like Olympus DAO (decentralized autonomous organization) (Chitra et al., 2022) are defined as futuristic or as Ponzi schemes (Thurman, 2021). In response to this, the Polkadot (Burdges et al., 2020) ecosystem launched Kusama (Burdges et al., 2020). Kusama is an experimental network with real economic incentives and a current market capitalization of around 300 million €. IOTA (Popov, 2018) was launched in 2016 as a temporary centralized protocol but has not yet switched to a decentralized one, as many features are still under development.

The aforementioned real-world examples demonstrate the importance and difficulties of migrating from well-established centralized solutions to experimental and decentralized ones. TPoW would allow for a smooth transition from a centralized to a decentralized system through a gradual approach: the system’s managers could monitor the impact of the decentralized features by not participating in the consensus and could regain control of the system in case any issues arise by leveraging their trapdoor knowledge.

Conclusion

This article introduced Trapdoor Proof of Work (TPoW), a mining algorithm that combines Proof of Work (PoW) and chameleon hash functions. By leveraging the knowledge of the trapdoor, trustworthy peers can efficiently generate blocks. From the perspective of potentially malicious peers, TPoW is as hard as the original PoW. Thus, TPoW enables fast probabilistic finality, low message complexity, horizontal scaling, and energy savings when the trusted party is online.

TPoW is applicable in systems composed of a trustworthy party and potentially malicious peers. TPoW-based consensus dynamically and automatically adapts to the evolving state of the system: decisions are delegated to the trustworthy party while it is online, whereas Byzantine fault tolerance is guaranteed when no trustworthy party is online. This behavior can be exploited to enhance service availability or facilitate the transition from a centralized to a decentralized protocol.

The main limitation of this study lies in its lack of testing in real-world scenarios with thousands of geographically distributed peers and real workloads. Future developments will aim to conduct additional tests to analyze TPoW’s performance in a more realistic environment, potentially through integration into Bitcoin’s codebase. Furthermore, a quantitative comparison of the energy wasted by PoW and TPoW could extend the analysis conducted in this study. Research efforts could also explore application-specific consensus algorithms or propose protocols for achieving consensus under alternative assumptions.

Supplemental Information

Supplemental Information 1 Golang Implementation of the TPoW protocol used for testing.

Click here for additional data file.

Additional Information and Declarations

Competing Interests

Author Contributions

Data Availability

1 Laptop model: Acer Nitro AN515-44. The test was performed using the implementation provided in Ref. (Willems, 2020).

The authors declare that they have no competing interests.

Vittorio Capocasale conceived and designed the experiments, performed the experiments, analyzed the data, performed the computation work, prepared figures and/or tables, authored or reviewed drafts of the article, and approved the final draft.

The following information was supplied regarding data availability:

The code is available in the Supplemental File.

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
