# Peer review of "Trapdoor proof of work"

_PeerJ Computer Science, doi:10.7717/peerj-cs.1815_

## Round 0.1 · original submission · Minor Revisions

Please pay close attention to the comments given by Reviewer 2 and Reviewer 3 and prepare a revision based on that.

Reviewer 1 ·

Basic reporting

The main content of the manuscript is that the algorithm based on the PoW constructed from chameleon hashes is more efficient in the presence of trusted parties, and the efficiency is basically the same as that of pow in the absence of trusted parties. TPoW can automatically switch to fast and slow algorithms in the presence or absence of trusted.

Experimental design

The main scheme does not clearly state the definition of a trusted party and the application scenarios for trusted parties, which is highly problematic.

Validity of the findings

The experiment was too brief, simply modeling the programmatic process, and did not use a clear description of the trusted party design of the experiment.

Cite this review as
Anonymous Reviewer (2024) Peer Review #1 of "Trapdoor proof of work (v0.1)". PeerJ Computer Science

·

Basic reporting

No Comment

Experimental design

No Comment

Validity of the findings

No Comment

Additional comments

This paper proposed a new consensus called "Trapdoor Proof of Work", a combination of Proof of Work and Chameleon Hash Function.
The paper sounds clear and unambiguous, with professional English used.
All literature references are up to date between the past 5 - 10 years
The figures and tables look very good and show results that are relevant to the hypothesis.

I need clarification on this:
In the Discussion Section, lines 309-210, it is written, "As expected, increasing the number of peers also increases the number of mined blocks, in conformity with PoW."

As far as I am concerned, the number of peers in a blockchain network does not directly increase the number of mined blocks. The rate at which blocks are mined is primarily determined by the difficulty of the cryptographic puzzle that miners must solve, which is adjusted to maintain a consistent block time and not by the number of miners or peers in the network.
In the context of blockchain and Proof-of-Work (PoW), "difficulty" refers to the measure of how difficult it is to find a hash below a given target. The Bitcoin network has a global block difficulty, which is adjusted every 2016 blocks, or approximately every two weeks, to maintain a target time of 10 minutes per block.

My Suggestion is:
Maybe you can add more references regarding your findings, and you can explain the "difficulty" you mention in your research.

Another suggestion is that you can add a comparison table or graphics between the original PoW and TPoW regarding the block-time generation; it will be better for your findings.

Cite this review as
Mardiansyah V (2024) Peer Review #2 of "Trapdoor proof of work (v0.1)". PeerJ Computer Science

Reviewer 3 ·

Basic reporting

Language:
It is suggested that a rigorous paper should avoid the use of the first person, please use “this paper/research” instead of “we/our”.

Article Structure
1. Abstract:
To make your abstract more effective, please consider to rewrite it with the following structure:
1) the overall purpose of the study
2) basic methodology of your research
3) major findings as a result of your analysis (Please state any quantitative findings from your experiment, i.e., number of mined blocks, energy efficiency produced)
4) a brief summary of your interpretations and conclusions.

2. Introduction
a. Clarity in Problem Statement: The paper identifies the scalability dilemma but could provide more clarity on how this dilemma affects blockchain systems. A brief explanation of why scalability is a challenge in the context of security and decentralization could enhance the introduction.
b. The problem statement could briefly discuss whether hybrid approaches (combining features of CFT and BFT) have been explored in the literature and why a dynamic approach is preferred in this context.

3. Background:
a. The proposed method needs to be compared with related methods in the literature. A Tabular comparison is suggested.

Experimental design

no comment

Validity of the findings

Assessment of Impact and Novelty: For Energy Efficiency, I suggest that the author describe further in Experimental Validation and Discussion to provde evidence of claim that says "TPoW consumes approximately 10% of the energy consumed by PoW". Furthermore, I believe joule will be a more appropriate metric to demonstrate energy usage rather than microsecond. Tools like Intel Running Average Power Limit (RAPL) can be used.

Additional comments

a. Risk Mitigation Strategy in Possible Applications: While the proposed solution addresses the inefficiency of BFT algorithms in the presence of trusted parties, it might be useful to briefly discuss or acknowledge potential risks associated with the dynamic approach. For instance, are there specific risks during the transition between CFT and BFT modes, and how are these mitigated?
b. State the limitation of this work and suggest how the limitations can be resolved in the future scope of the work.

Cite this review as
Anonymous Reviewer (2024) Peer Review #3 of "Trapdoor proof of work (v0.1)". PeerJ Computer Science

---

## Round 0.2 · accepted · Accept

Thank you for addressing reviewer comments. I have assessed the revision and I am happy with the changes made. Please go over the manuscript in full detail and prepare the final ready-for-publication version.